# The Multi-Scale Layering-Structure of Thermal Microscale Profiles

## Andrew Folkard

Lancaster Environment Centre, Lancaster University, Lancaster LA1 4YQ, UK; a.folkard@lancaster.ac.uk

**Abstract:** Thermal microstructure profiling is an established technique for investigating turbulent mixing and stratification in lakes and oceans. However, it provides only quasi-instantaneous, 1-D snapshots. Other approaches to measuring these phenomena exist, but each has logistic and/or quality weaknesses. Hence, turbulent mixing and stratification processes remain greatly under-sampled. This paper contributes to addressing this problem by presenting a novel analysis of thermal microstructure profiles, focusing on their multi-scale stratification structure. Profiles taken in two small lakes using a Self-Contained Automated Micro-Profiler (SCAMP) were analysed. For each profile, buoyancy frequency ($N$), Thorpe scales ($L_T$), and the coefficient of vertical turbulent diffusivity ($K_Z$) were determined. To characterize the multi-scale stratification, profiles of $d^2T/dz^2$ at a spectrum of scales were calculated and the number of turning points in them counted. Plotting these counts against the scale gave pseudo-spectra, which were characterized by the index $D$ of their power law regression lines. Scale-dependent correlations of $D$ with $N$, $L_T$ and $K_Z$ were found, and suggest that this approach may be useful for providing alternative estimates of the efficiency of turbulent mixing and measures of longer-term averages of $K_Z$ than current methods provide. Testing these potential uses will require comparison of field measurements of $D$ with time-integrated $K_Z$ values and numerical simulations.

**Keywords:** fractal; lakes; mixing; multi-scale; stratification; turbulence

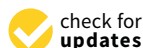



## 1. Introduction

Stratification and mixing of lakes and oceans govern vertical fluxes of dissolved and particulate matter—including nutrients, pollutants and planktonic biota—and therefore are of great importance for understanding chemical and biological aspects of surface waterbodies [1]. They are also fundamentally important processes for the global heat energy budget [2]. Their actions lead to lakes and oceans having vertical density profiles that can be divided into distinct layers. At the simplest, macroscale level, these include, in oceans (lakes), a surface mixed layer (epilimnion), thermocline (metalimnion) and deeper, generally more quiescent layers (hypolimnion). This macroscale, three-layer structure is the classic example of a relatively strongly-stratified layer being found between two relatively well-mixed layers. The development of high spatial-resolution microstructure profilers in the last decades of the 20th century led to the discovery that increasingly subtle forms of this layering structure occurred at ever finer scales (Figure 1). Notwithstanding its subtlety, this finer-scale layering is significant because even small changes in density can have significant effects on vertical fluxes of plankton and dissolved and particulate materials [3].

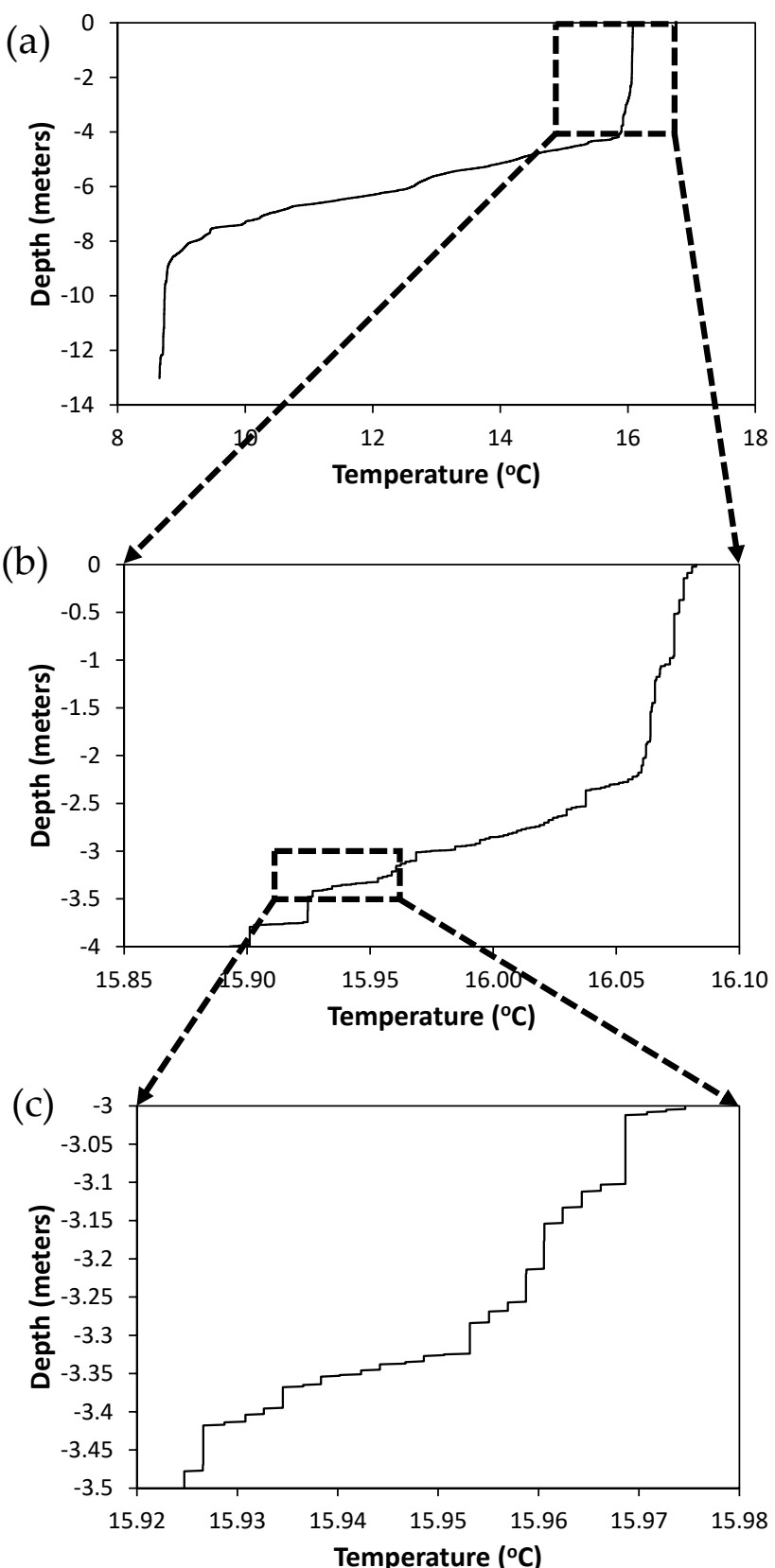

**Figure 1.** Example of the multi-scale layering structure in thermal microstructure profiles (profile recorded in Blelham Tarn at 11:00 on 11 August 2008): (**a**) full-depth profile; (**b**) zoomed into top 4 m; (**c**) zoomed into 3–3.5 m below surface. See Figure 2 and associated text for explanation of the "chunkiness" in the fine structure shown here.

A wide variety of microstructure profilers have been developed which have revealed fluctuations in temperature, velocity shear, conductivity, chlorophyll, turbidity and chemical concentrations at the scale of the smallest turbulent eddies: centimeter- or millimeter-scale [4,5]. Specifically, thermal microstructure profiling has been an established technique for investigating mixing and stratification in lakes and oceans for many years [2,6], and continues to be widely used [7–9]. In freshwater lakes—the object of the study reported here—the use of thermal profilers is particularly useful, because in the absence of salinity variations, density (and therefore buoyancy and the transport and mixing it induces) is primarily determined by temperature alone. Data from these thermal microstructure profilers reveal not only the layered stratification structure of the water column in great detail, they also capture snapshots of the turbulent stirring of the profile, which disturb it from a monotonic state. Length scales of the turbulent overturns causing this stirring are most commonly quantified as Thorpe scales $L_T$. These are calculated from Thorpe displacements $\delta_T$ (differences in height of temperature records between raw, unsorted and monotonically-sorted profiles) as $L_T = <\delta_T^2>^{1/2}$ i.e., the root mean square value [10]. The presence of these overturns is used to identify actively mixing layers, while the stratification structure of the profile can be used to identify already mixed layers (see [11] for discussion of the importance of this distinction). The edges of active turbulent layers are often defined as the point where $\delta_T$ falls to zero [12,13]; hereinafter this is referred to as the "$\delta_T = 0$" method of mixing layer identification. The statistics of stratification and mixing in these layers, especially the surface mixed layer, are important parameters for a wide variety of ocean and lake phenomena, including biological productivity, air-water exchange processes, and long-term climate change [14,15].

Thermal microstructure is also widely used to derive values of the coefficient of vertical turbulent diffusivity, $K_Z$. These may then be used to quantify the rate of vertical turbulent fluxes [16], which are essential parameters in models of lake heat budgets, nutrient cycling, plankton population dynamics and many other aspects of surface waterbodies. When deriving $K_Z$ from thermal microscale profiles, it is most commonly calculated using a method commonly known as the "Batchelor method", after its creator [17]. This entails transforming the temperature profile into a temperature gradient spectrum and then obtaining the value of the rate of turbulent kinetic energy dissipation ($\varepsilon$) by fitting the theoretical Batchelor spectrum [17] to that temperature gradient spectrum at high wavenumbers [18]. $K_Z$ is then calculated as

$$K_Z = \Gamma \varepsilon / N^2 \tag{1}$$

Here, N is the buoyancy frequency, $N = (g(\partial \rho / \partial z)/\rho_0)^{1/2}$, and the parameter $\Gamma$ is defined as

$$\Gamma = R_f / (1 + R_f) \tag{2}$$

where $R_f$ is the flux Richardson number, which measures the efficiency of the mixing process. In physical terms, this can be thought of as the proportion of turbulent kinetic energy that is converted into irreversible changes in the potential energy of the profile, rather than being dissipated down the turbulent energy cascade. In this sense, it can be written as $Rf = b/(b + \varepsilon)$ where b represents buoyancy flux (i.e., changes in potential energy) and $\varepsilon$ is turbulent dissipation. Values of vertical diffusivity calculated using the Batchelor method have also been compared with those calculated assuming a direct relationship between $K_Z$ and the Thorpe scale $L_T$. Using the parameterization $K_Z = 2\nu(\varepsilon/\nu N^2)^{1/2}$ for the energetic turbulent regime $\varepsilon/\nu N^2 > 100$, this relationship can be written as [19,20]:

$$K_Z = 1.6\nu^{1/2}L_T N^{1/2} \tag{3}$$

This method is referred to hereinafter as the "Thorpe scale method".

A significant problem with microstructure measurements, and therefore the mixing and stratification parameters derived from them, remains that they reflect quasi-instantaneous, 1-D snapshots of the turbulence field at specific locations. Other approaches to measuring $K_Z$, such as eddy correlation or tracer diffusion techniques [21,22] provide temporal averages of the vertical flux effects of the turbulence, but are much more time-consuming and logistically difficult to set up, and still only provide data on a very small spatial and temporal scale compared to that required to understand the global ocean-scale, or even whole-lake scale, effects of turbulent mixing. This is because of the great intermittency in time and space of turbulent activity, which means that sampling of turbulent activity and thus its effective parameterisation for use in predictive models is highly problematic. To date, the rate and density of sampling of these turbulent events is far too low to give us a clear picture of their distribution and global characteristics [23]. This is also true for lakes, where most information of vertical mixing in lake thermoclines is based on laboratory measurements and simulations [22], as it is for oceans.

As a result, the distribution (in space and time) of turbulent mixing and stratification processes, and their characterization in terms of parameters such as $L_T$ and $K_Z$ continue to be widely-studied [24–27]. A novel approach to this problem has been proposed by [28], who adopted a statistical physics perspective and characterised the efficiency (increase in potential energy to total energy input ratio) as a distinction between changes in the coarse-grained buoyancy profile, which represents the irreversible increase in potential energy, to the remaining energy, which is lost to fine scale fluctuations of velocity and buoyancy. They found that the variation of mixing efficiency with the Richardson number strongly depended on the background buoyancy profile, and that the mixing efficient has a maximum value of 0.25, which agreed well with predictions based on the more usual kinematics-based approach. This is consistent with measurements from the field [29] and laboratory [30].

This perspective suggests that considering the scale spectrum of layers in the microscale temperature profile might provide a way of understanding how both the currently active turbulence, and that which created the multi-scale layering of the temperature profile prior to its recording, might provide insights regarding the parameterisation of the turbulent mixing process that is required for it to be robustly incorporated into lake and ocean models. Therefore, given that this aspect of microstructure profiles does not appear to have been considered in the literature previously, this paper aims to:

- investigate the layered structure of microscale temperature profiles;
- identify its essential properties and determine whether any of them are universal;
- determine whether they vary consistently in relation to other parameters; assess whether (and how) they can be interpreted as a diagnostic tool for understanding turbulence mixing processes and their consequences better.

## 2. Materials and Methods

### 2.1. Site Description and Data Collection

To explore the multi-scale layering structure concept described above, thermal microstructure profiles taken in two lakes, Esthwaite Water and Blelham Tarn, during the summer stratified period of 2008 using a Self-Contained Automated Micro-Profiler (SCAMP, Precision Measurement Engineering Inc., San Diego, CA, USA) were used. This data does not have any characteristics that distinguish it from microstructure profile data taken in any other waterbodies, it was chosen only because it was readily available. Both lakes are small, glacially-scoured and lie within the catchment of Windermere in the Lake District of Northwest England. The larger of the two, Esthwaite Water (54.36° N, 2.99° W), has a surface area of 0.96 km$^2$, a total volume of $6.7 \times 10^6$ m$^3$ and a mean depth of 6.9 m [31]. Blelham Tarn (54.40° N, 2.98° W) has a surface area of 0.1 km$^2$ and a mean depth of 6.8 m [32]. Thus, they are of similar depth, but Esthwaite Water has a surface area approximately ten times that of Blelham Tarn.

SCAMP was operated in upward-looking mode (see, for example, [33]). It was deployed from a boat with weights and a baffle attached, which caused it to drop through the water column at an angle of approximately 45° to the vertical. When it reached a user-defined depth, a pressure sensor caused a screw to turn, which released the weights. This made the instrument positively buoyant, so that it rose vertically through water thus undisturbed by its descent, at a speed of approximately 0.1 ms$^{-1}$, recording temperature and depth (pressure) at 100 Hz (thus generating data points with approximately 1 mm spatial resolution). Once it reached the surface, the weights were recovered and re-attached, the pressure sensor re-set and the next profile initiated.

The analysed profiles were recorded on eight days in Esthwaite Water (22 May, 16 June, 21 July, 31 July, 4 August, 1 September, 2 October and 3 October) and four days in Blelham Tarn (9 June, 23 June, 28 July and 11 August). All of these days fell within the summer stratified period for the lakes, which runs from onset in March or April to turnover in October. The profiles were all recorded during the daytime, between approximately 9:30 a.m. and 5:00 p.m., the time for each day being dependent on logistical arrangements. On each day, a group of six profiles were recorded, separated by between five and ten min, thus covering a total of 30 to 60 min in total. All measurements were taken at the deepest point in each lake, the profiles having maximum depths of approximately 14 m.

### 2.2. Data Processing

Each profile was converted from its raw form into ASCII format files using processing software provided by the manufacturer. The profiles were then truncated at the bottom, where data recording had begun before the SCAMP's upward travel had started, and at the top, where recording had continued after it had breached the surface. The truncated profiles thus began at the deepest point recorded in the raw profiles, and ended where the (pre-calibrated) pressure measurements indicated zero depth.

As the mm-resolution of the raw profiles was only approximate, the truncated profiles were then interpolated to exactly 1 mm-depth resolution using an inverse distance-weighted mean of all recorded temperature data within 5 mm of each point on the 1 mm-resolution scale. To remove noise from the data, and thus prevent false identification of turbulent overturns, the de-noising method of [34] was then applied to each profile, following [19]. In the original version of this method, the noise threshold is defined in terms of the density. Since the data used here was temperature data, the density threshold needed to be converted to a temperature threshold. However, in standard practice, density is calculated from temperature using a fifth order polynomial [35] and inversion of fifth order polynomials is intractable. To convert the previously-used density noise threshold to a temperature noise threshold, therefore, a linear fit of the density-temperature relationship in the range 8–22 °C (where all of our temperature data lay) was performed, and the gradient of this was used to convert the density threshold of [34] ($5 \times 10^{-4}$ kgm$^{-3}$) to the temperature threshold of $3.5 \times 10^{-3}$ °C, which was used in this cleaning process. The results of both the interpolation and noise removal processes are illustrated in Figure 2. At the relatively fine scale indicated in this figure, the "cleaned" (noise removed) profile may appear chunky, and possibly artificial, at first sight. However, this is simply the result of cleaning the profile to appropriate spatial and temperature resolutions of 1 mm and $3.5 \times 10^{-3}$ °C, respectively, following [19,34]. The profiles thus cleaned are those which are analysed as described below.

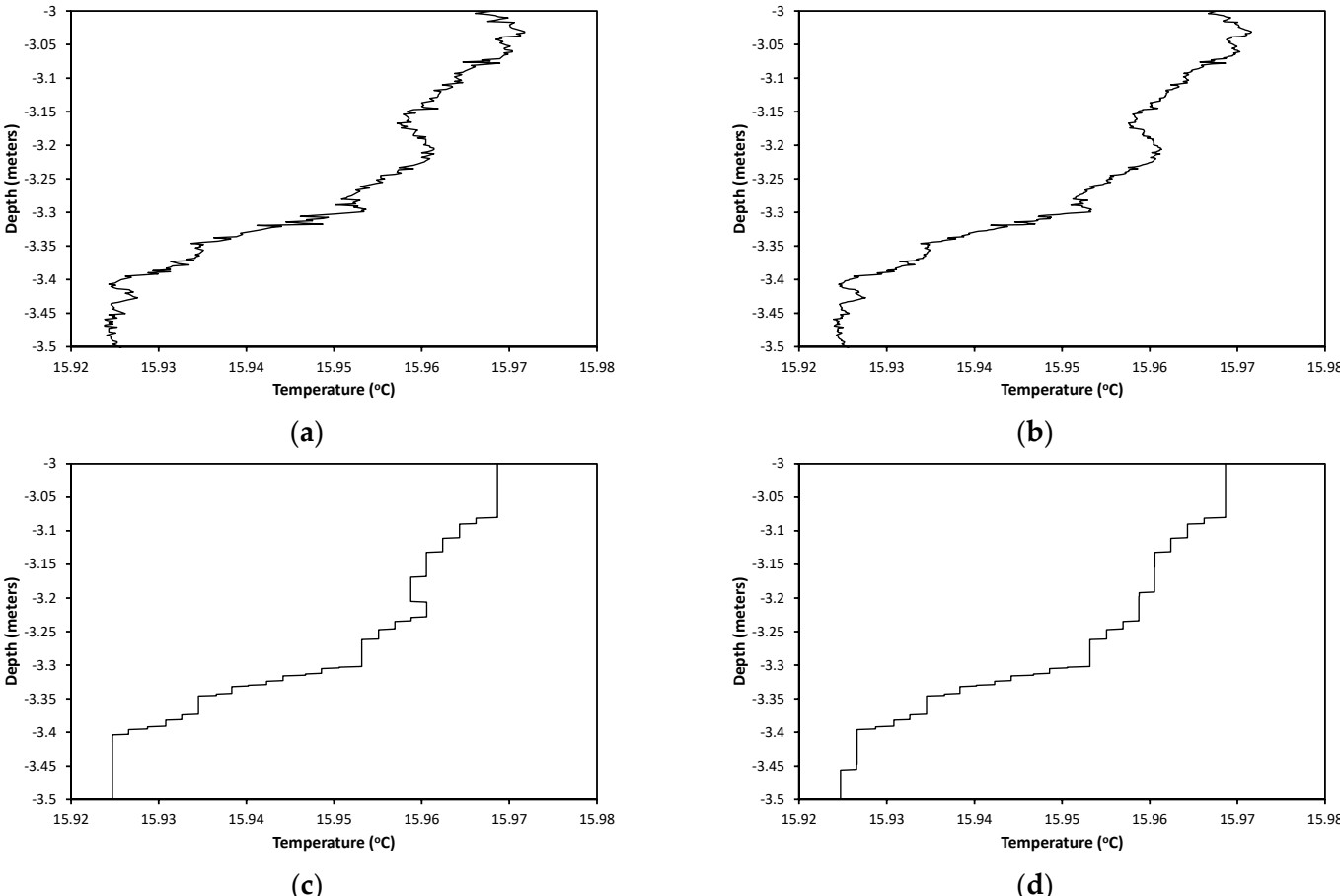

**Figure 2.** Examples of profile segments illustrating the effects of interpolation to 1 mm vertical spatial resolution and temperature resolution of $3.5 \times 10^{-3}$ °C: (**a**) raw profile; (**b**) spatially interpolated profile; (**c**) spatially and temperature interpolated profile; (**d**) Thorpe-ordered version of (**c**). Profile recorded in Blelham Tarn at 11:00 on 11 August 2008).

### 2.3. Identification of Mixing Layers and Thorpe Scales

The Thorpe scale at each depth in each mm-resolution profile was calculated from a 0.5-m depth window centered on that depth (truncated by the start or end of the profile for points within 0.25 m of the top or bottom of the profile). These were converted to centered Thorpe displacements for the purpose of illustration using the method of [36].

To identify and delineate actively mixing patches within the profiles, a slightly different approach to the $\delta_T = 0$ method was adopted. Each temperature-depth profile was sorted by temperature (as one would to calculate Thorpe displacements), then cumulatively summed via the (re-sorted) depths of this profile, and at each depth this sum was compared with the cumulative sum of the unsorted (i.e., monotonically-ordered) depth profile. The point where these two sums are first equal (working from the top downwards) is the point above which the mixing is entirely 'self-contained': all of the points in this region in the sorted profile are also within it in the unsorted profile, and no points from outside it have been moved into it by the sorting process. This is defined as the uppermost mixing patch. Since the sums at this point are equal, the process re-sets and the next point down at which the sums again become equal marks the bottom of the next mixing patch down. In theory, this is a more robust method of patch identification than the $\delta_T = 0$ method, since it avoids the possibility of a point within the sorted profile being (coincidentally) at the same point in the unsorted profile (which would give $\delta_T = 0$ at that depth) but being surrounded by points that have been mixed upwards and downwards across it (i.e., being within a mixing patch, not at its edge). In practice, however, this method segmented the profiles in a manner that was indistinguishable by visual inspection from the segmentation done using

the $\delta_T = 0$ method. Nevertheless, it was used hereinafter and is presented as a very simple, and potentially more robust, method of profile segmentation into distinct mixing patches.

### 2.4. Calculation of $K_Z$

The Batchelor method of calculating $K_Z$ was applied to each of the six profiles from each date, using software incorporating this method provided by SCAMP's manufacturer, PME Inc. Although there has been much discussion of the variability of $R_f$ and $\Gamma$ in the literature [13,37], the standard approach of assigning a value of $\Gamma = 0.2$ was taken. Mean and standard deviation values of $K_Z$ were calculated for each 0.5-m depth bin from the surface downwards and plotted against the center point of each bin (0.25 m, 0.75 m etc.). The Thorpe scale method was then used to calculate $K_Z$ at every depth (i.e., at millimeter resolution) in the profile using 0.5-m centered windows as described above for $L_T$, calculating the dynamic viscosity, $\nu$, as a function of temperature. Mean and standard deviation values of these Thorpe-scale derived $K_Z$ values were then calculated at the same depths used for the Batchelor method (i.e., 0.25 m, 0.75 m etc.).

### 2.5. Layer Structure Analysis

Analysis of the multi-scale layering of the Thorpe-ordered profiles was carried out using best-fit straight lines ("rulers") of lengths from 3 mm to the full profile length (Figure 3). For the 3 mm case, a straight line was first fitted to the first, second and third points in the profile, and its gradient (dT/dz) assigned to the center point of this set (i.e., the depth of the second point). This was repeated for the second, third and fourth points, assigning the gradient to the depth of the third point, and so on down to the bottom of the profile, the last gradient value being assigned to the penultimate depth point in the profile. The second derivative ($d^2T/dz^2$) profile was then calculated from this set of dT/dz values, using the same 3-mm spatial scale. This process was then repeated using a 5-mm spatial scale, and so on, up to the largest odd number less than the full length of the profile (only odd numbers were used so that the center point of each fit line corresponded to a specific depth in the profile). Thus, a full set of $d^2T/dz^2$ profiles at a spectrum of different spatial scales was obtained.

At depths where $d^2T/dz^2$ peaks, the temperature profile is changing most rapidly from a low-gradient (well-mixed) section to a high-gradient (highly-stratified) section; where there are troughs in $d^2T/dz^2$, the temperature profile is changing most rapidly the other way. Thus, these points identify "shoulders" in the temperature profile that distinguish relatively-mixed layers from relatively-stratified ones (Figure 4). The total number of these shoulders was calculated for each ruler-length, and a pseudo-spectrum (ruler-length, or scale S vs. number of shoulders or layers $N_L$) ws then plotted for each two-meter depth bin from the surface down to 14 m. These pseudo-spectra were generally closely fitted by power law regression lines, implying a fractal structure. Therefore, the fractal dimension, D, was calculated for each profile such that $N_L = aS^{-D}$. As well as the fractal dimension of the full spectrum, separate values of D for the fine, intermediate and coarse-scale sections of the spectrum were calculated. Correlations of all of these forms of D with N, $L_T$ and $K_Z$ were investigated using Pearson's product-moment correlation coefficients and their associated statistical significance (*p*-values).

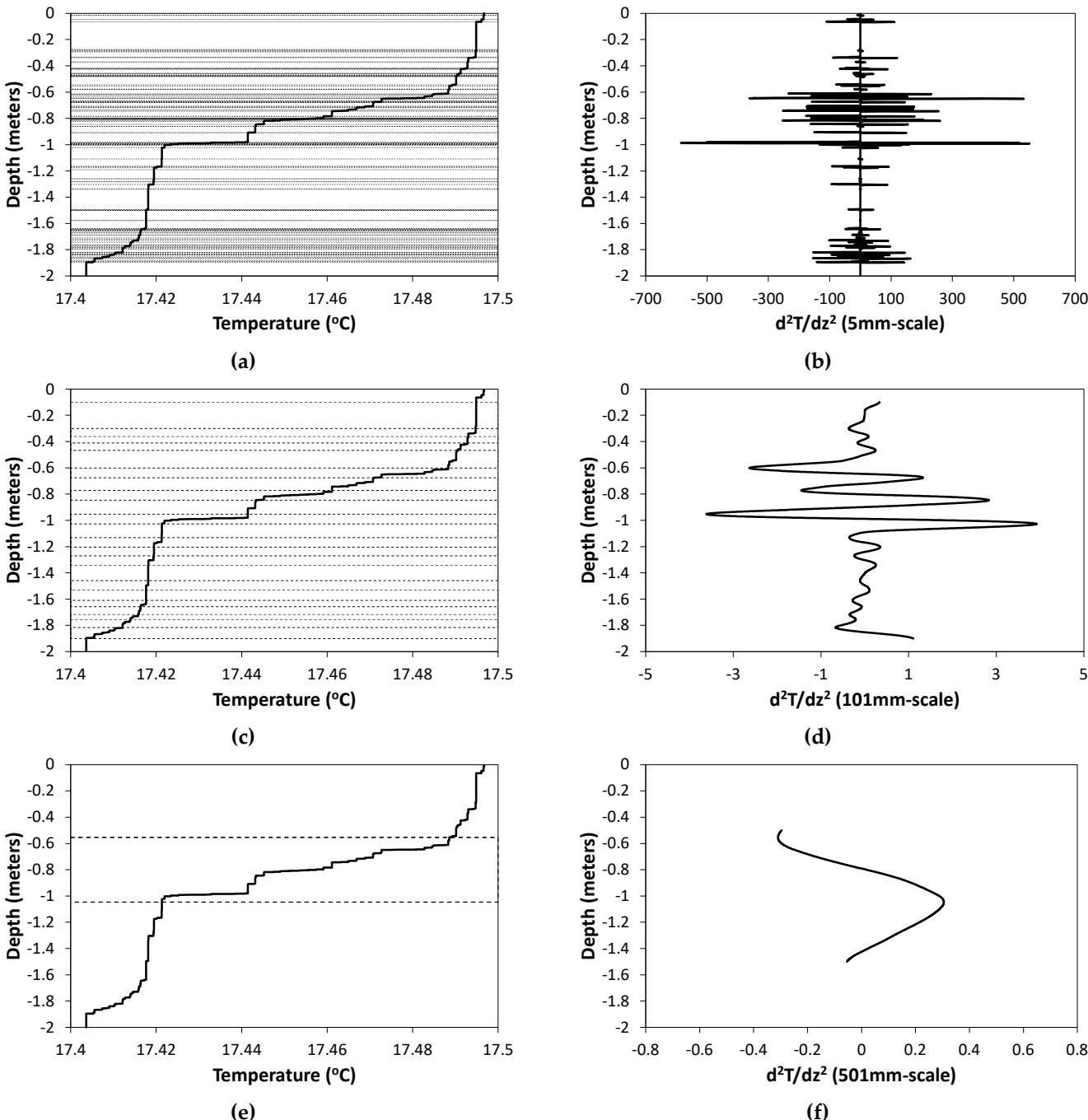

**Figure 3.** Illustration of the process of calculating the number of layers identified at different spatial scales, by counting turning points (peaks and troughs) in the $d^2T/dz^2$ profile: (**a**) location of layer edges ("shoulders") at 5 mm scale; (**b**) $d^2T/dz^2$ at 5 mm scale (**c**) location of layer edges at 101 mm scale; (**d**) $d^2T/dz^2$ at 101 mm scale; (**e**) location of shoulders at 501 mm scale; (**f**) $d^2T/dz^2$ at 501 mm scale.

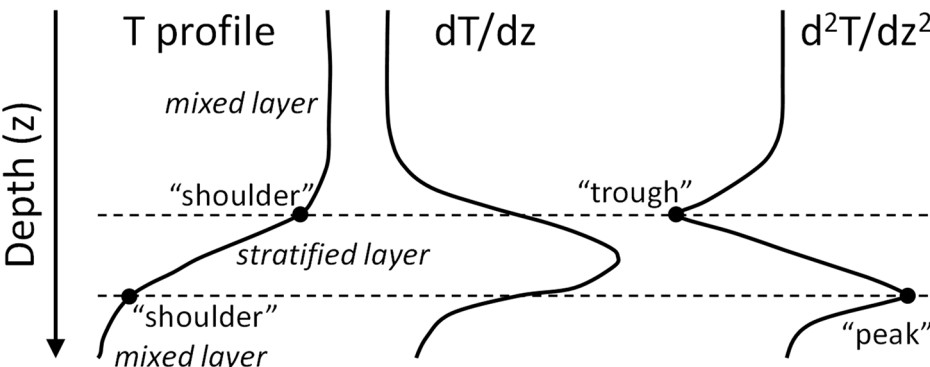

**Figure 4.** Sketch showing how troughs and peaks in the $d^2T/dz^2$ profile identify boundaries between alternately (relatively) strongly- and weakly-stratified layers in the temperature profile.

## 3. Results

### 3.1. Mixing and Stratification Structure

SCAMP temperature profiles from each of the twelve data collection days are presented in the left-hand panels of Figure 5 (Esthwaite Water) and Figure 6 (Blelham Tarn). These are cleaned, individual, unsorted profiles (rather than averages of each set of six). The segmentation of the profiles into separate mixing layers also provide a clear visual illustration of the proportion of the profile that is actively mixing at the time of each profile, and the locations of the mixing and non-mixing regions. This is also indicated by the centered Thorpe displacements, showing the coincidence of the mixing layers identified by the method used here and the $\delta_T = 0$ method. The seasonal timescale transition through a maximally-stratified state (e.g., 4 August) to a deepening surface mixed layer as overturn approaches (e.g., 3 October) can be seen clearly, especially in Esthwaite Water (Figure 5). Diurnal timescale changes can also be seen clearly by comparing the 2 and 3 October profiles in Figure 5. On 2 October, the profiles were taken in the afternoon, and the surface layers shows some stratification and a relatively quiescent state in terms of Thorpe displacements. On 3 October, the profiles were taken in the morning, the surface layers is much more completely mixed (presumably due to overnight convection) and much more actively stirring, as indicated by the much larger Thorpe displacements.

### 3.2. Comparison of $K_Z$ Values

Plots of $K_Z$ values for 0.5-m depth bins calculated using both the Batchelor and Thorpe scale methods are shown in the right-hand panels of Figure 5 (Esthwaite Water) and Figure 6 (Blelham Tarn). In general, the plots have the expected structure for a stratified lake, with relatively high $K_Z$ values in the surface mixing layer and at the bottom of the profile in the near-bed layer, and lower values in the "quiet interior" at mid-depths [6]. In many places, the $K_Z$-values from the two methods agree very well. This agreement appears stronger in the surface layer, particularly later in the year in Esthwaite Water, and stronger in general in Blelham Tarn than in Esthwaite Water. In many other places, however, the agreement is poor. This is particularly the case at depth (and particularly in Esthwaite Water), where the Batchelor method values are generally larger than the Thorpe-scale method values, by at least an order of magnitude (and in some places several orders of magnitude) and are also as large or larger than the surface layer values.

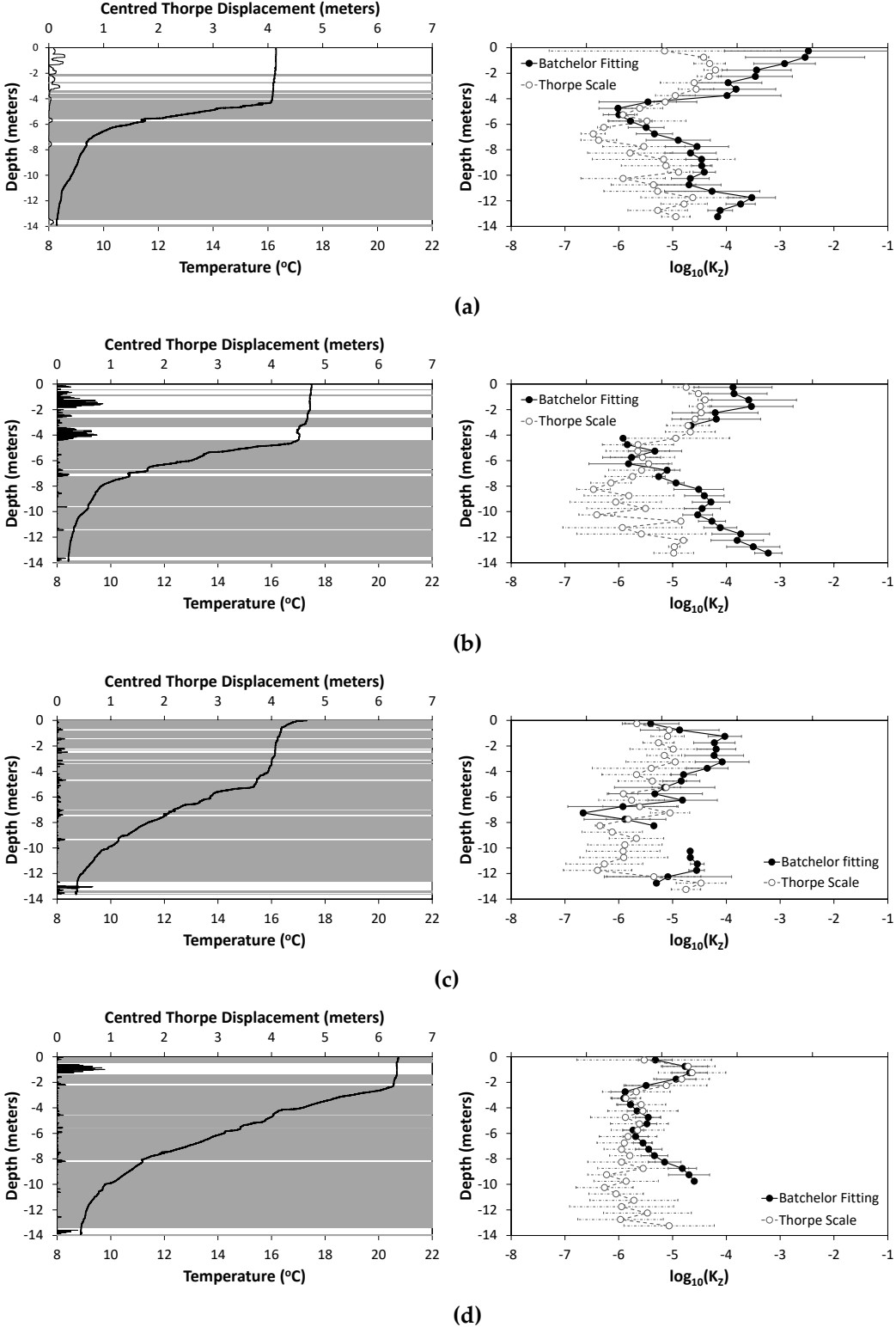

**Figure 5.** *Cont.*

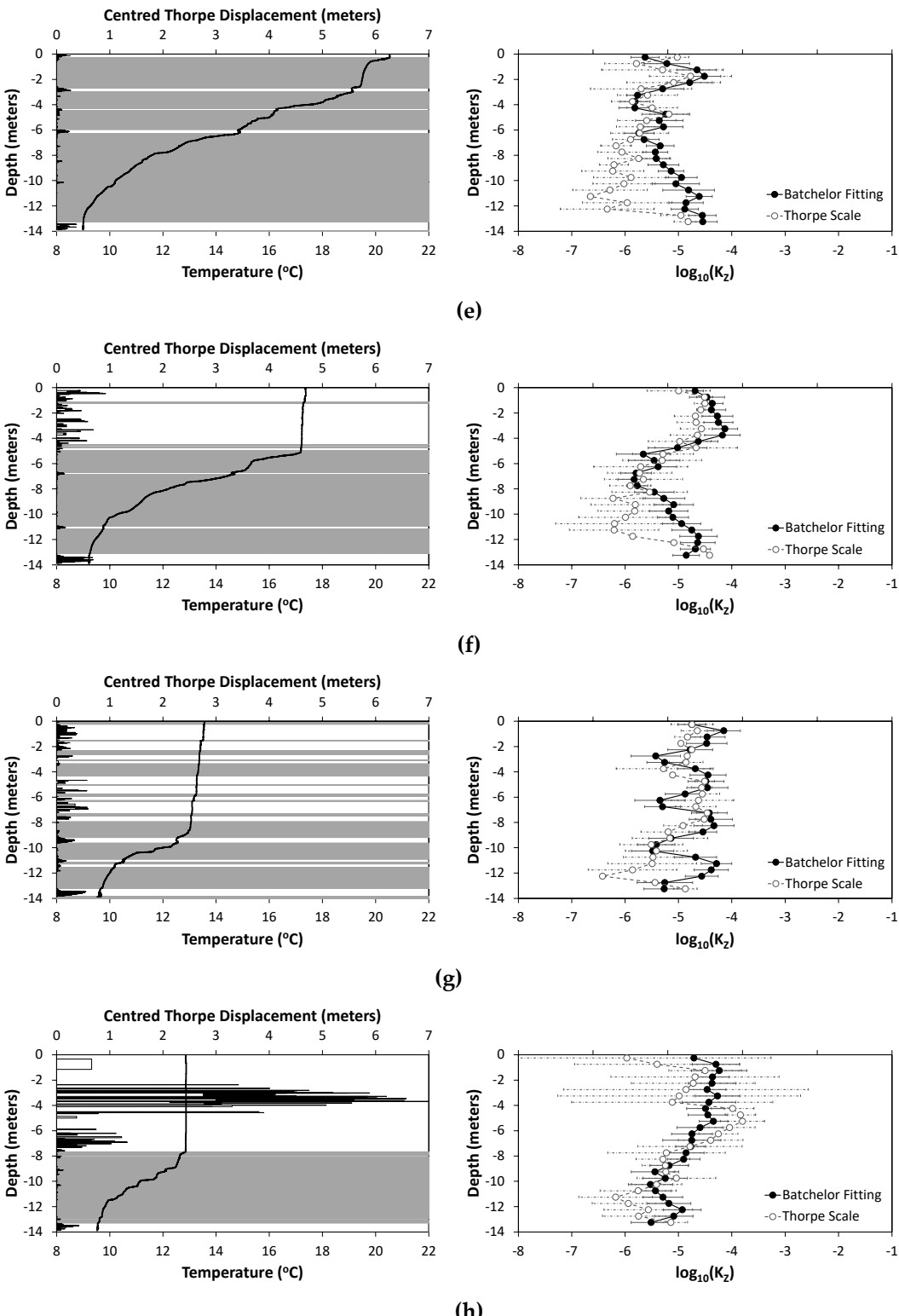

**Figure 5.** Plots showing (in left hand panels) temperature profile (thick black line), mixing layer segmentation (horizontal gray lines) and centered Thorpe displacements (black line at left hand side); and (in right hand panels) values of the coefficient of turbulent diffusivity ($K_Z$) using the two methods described in the text, for all sampling dates for Esthwaite Water. Profiles are individual examples from the set of six profiles taken on each date, recorded at (**a**) 16:21 on 22 May; (**b**) 12:33 on 16 June; (**c**) 12:15 on 21 July; (**d**) 11:25 on 31 July. (**e**) 12:07 on 4 August; (**f**) 12:38 on 1 September; (**g**) 16:05 on 2 October; (**h**) 09:56 on 3 October.

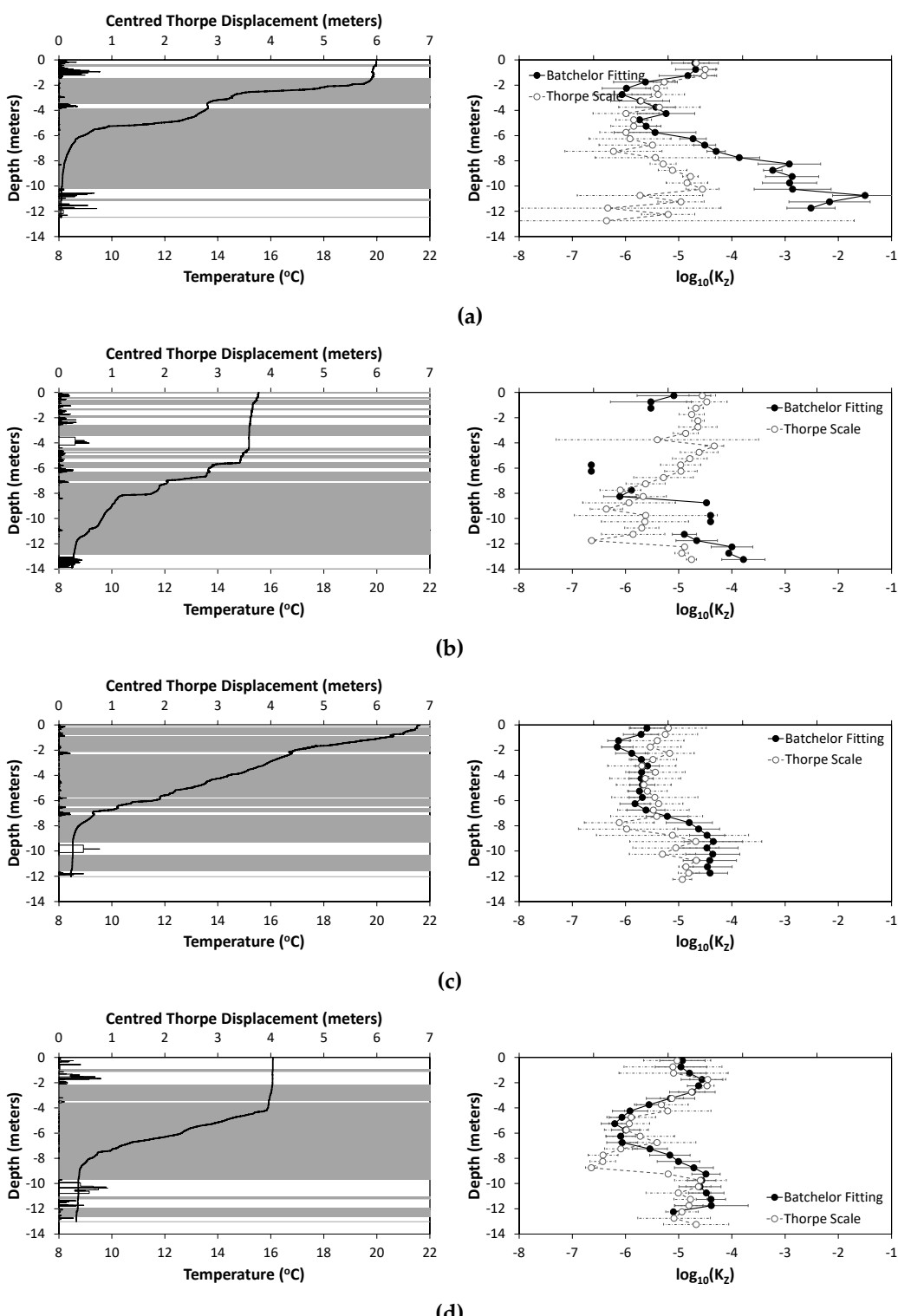

**Figure 6.** As for Figure 5, for Blelham Tarn. Profiles recorded at (**a**) 13:33 on 9 June; (**b**) 12:41 on 23 June; (**c**) 11:07 on 28 July; (**d**) 11:00 on 11 August.

### 3.3. Layer Structure Pseudo-Spectra

Figure 7a,b show two contrasting examples of layer structure pseudo-spectra, with power law lines fitted, taken from individual depth bins of the same profile (the one taken at 12:28 on 16 June). The equations of the best-fit power law lines are shown on these plots: the power index, which, as explained above, is denoted D (c.f. fractal dimension)

is used to quantify the gradient of the dataset. Figure 7c shows the pseudo-spectra from all depth bins of this example profile together. Figure 7a illustrates a fractal structure across all scales (i.e., the gradient is much the same across the whole range of data values), whereas in Figure 7b the fitted power law line digresses strongly from the data, especially at smaller scales. The shapes of these pseudo-spectra are interpreted as follows: where they are steeper, the number of extras layers identified as the scale is reduced is higher, i.e., there is a lot of layering-structure at that particular scale; where they are flatter, there is little layering structure. Thus, in Figure 7b, the flatness of the data at smaller scales (say, <20 mm) implies that there are very few layers smaller than 20 mm in size, i.e., there is very little fine-scale structure in the profile.

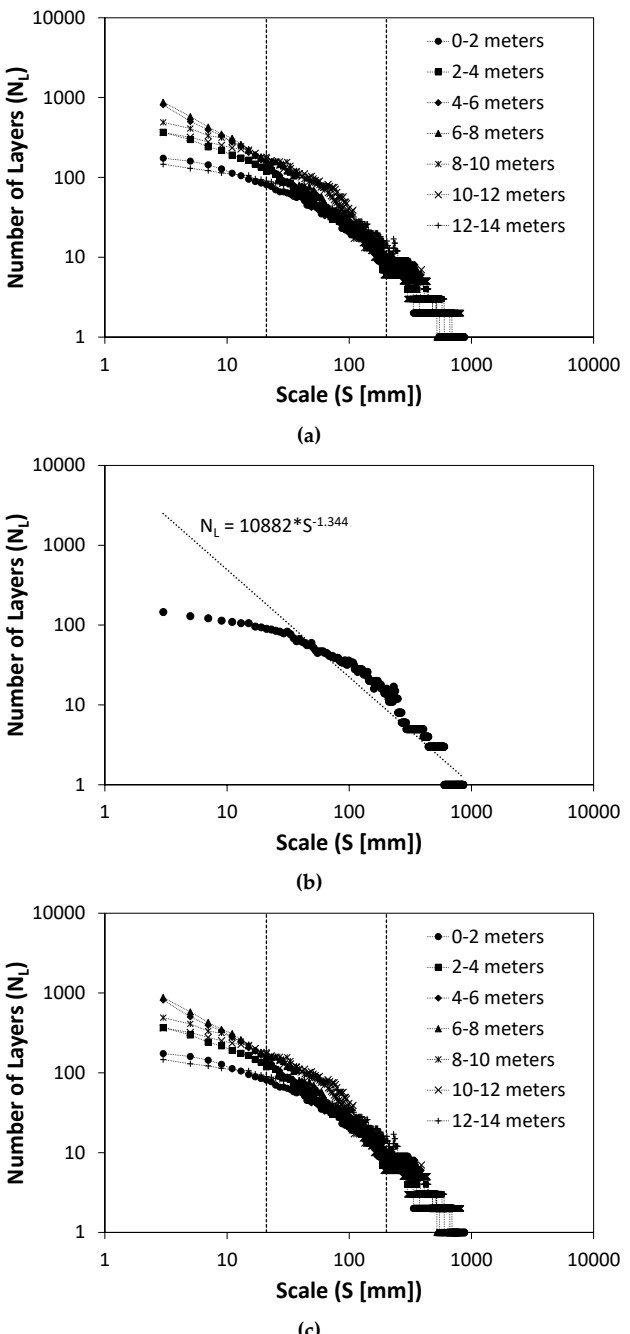

**Figure 7.** Examples of pseudo-spectra of the layering structure, all for the profile recorded at 12:28 on 16 June 2008 in Esthwaite Water. (**a**) depth bin from 4 to 6 m; (**b**) depth bin from 12 to 14 m; (**c**) all 2 m-depth bins together.

Following this interpretative approach, the gradient of the best fit power law line to the whole data set in each plot can be said to give a quantitative indication of the overall level of layering structure in the temperature profile. To gain insights into the amount of layering structure at different spatial scales, each spectrum is divided into three spatial scales, which are referred to hereinafter as "fine", "intermediate" and "coarse", and best-fit power law lines are calculated for each one. This division is based on a subjective judgment of where the breaks in the gradient of the spectra tend to fall, derived from visual inspection of spectra from all depth bins of all profiles. Thus, the fine-scale is defined as <20 mm, the intermediate scale as >20 mm but <200 mm, and the coarse scale as >200 mm. These divisions are shown in Figure 7c. The gradients for each of these scales are denoted $D_F$, $D_I$ and $D_C$, respectively.

### 3.4. Variation of D with Depth, between Lakes and over Time

To explore the variation of the fractal dimension of the profiles, their variations with depth, between lakes and over seasonal and diel timescales are first considered. Differences in the mean values of D in each 2-m depth bin between the two lakes are shown in Figure 8. The mean values and standard deviations in this plot have been calculated from data from all profiles, thus seasonal variations are both hidden in, and serve to smear out, the standard deviation bars. The value of D peaks deeper in Esthwaite Water than in Blelham Tarn, and is smaller at the bottom of Blelham Tarn than in Esthwaite Water. But overall, both lakes show a similar pattern of lower values at the surface and bottom, and higher values at mid-depths. This suggests an association between less layering structure (lower values of D) and more weakly-stratified and mixed (or mixing) parts of the profiles.

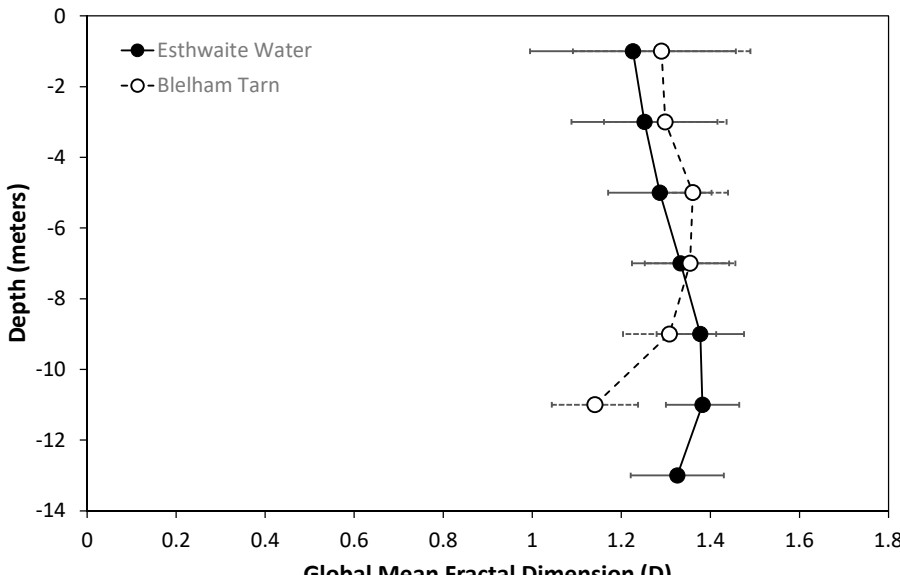

**Figure 8.** Mean (± one standard deviation) values of D in each 2 m-depth bin, averaged over all profiles from all dates in each lake.

The variation in profiles of D over seasonal and diel timescales is illustrated in Figure 9. Again, the very weakly-stratified, strongly mixing upper part of the 3 October profile is characterised by smaller values of D, indicating less layering structure, while the values of D for 2 October are higher, and those for 4 August are higher still.

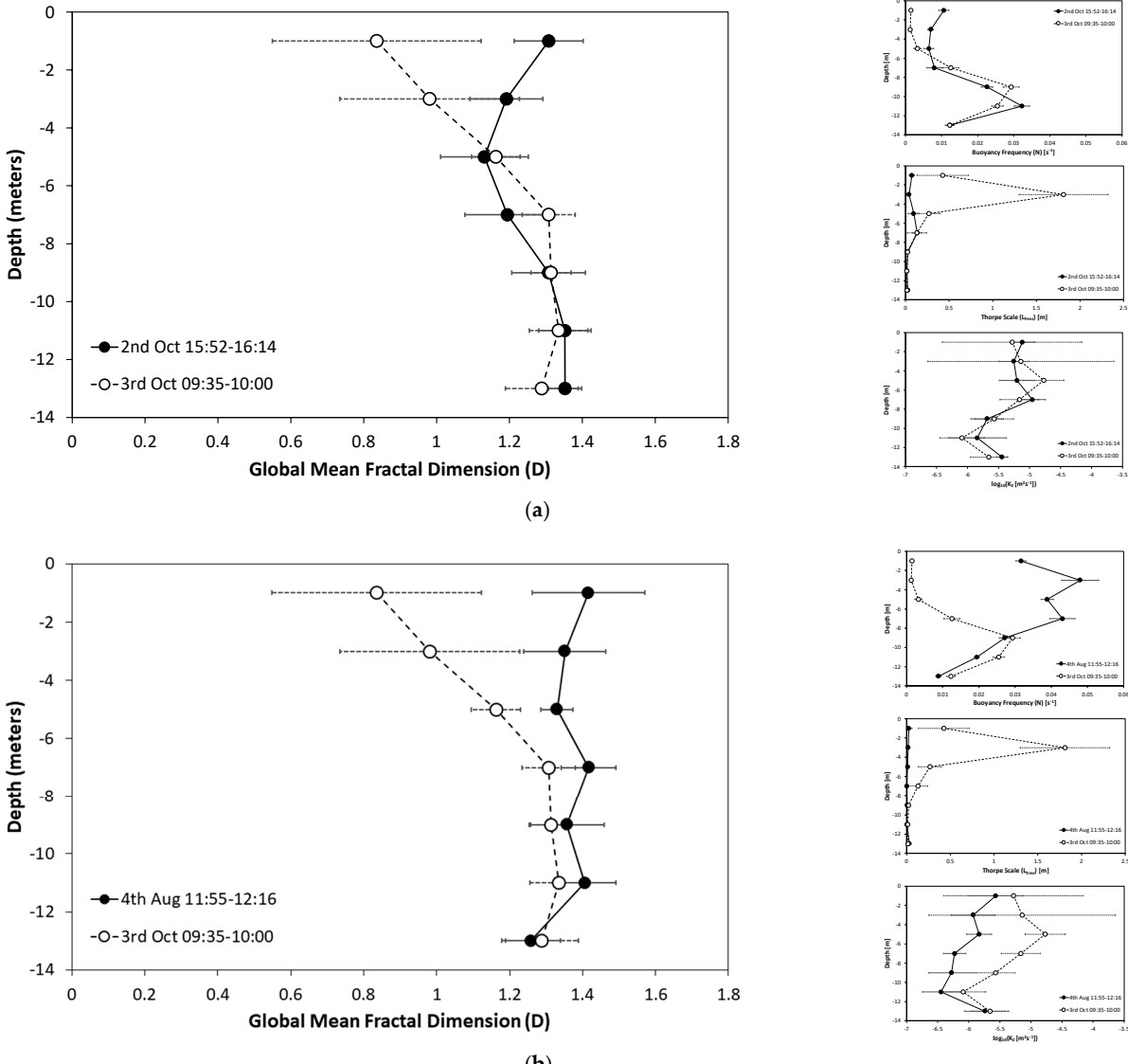

**Figure 9.** Comparisons of mean ($\pm$ one standard deviation) values of D in each 2 m-depth bin (left hand panels) for profiles separated by (**a**) daily variations and (**b**) seasonal variations. Right hand panels show buoyancy frequency (top), root mean square Thorpe scale (middle) and $\log_{10}(K_Z)$ (bottom) for each date, for comparison.

### 3.5. Variation of D with N, $L_T$ and $K_Z$

To explore further the way in which D varies, it and its scale-specific forms $D_F$, $D_I$ and $D_C$ were plotted against the three standard parameters which quantify stratification and mixing—the buoyancy frequency N, the Thorpe scale $L_T$ and the coefficient of vertical turbulent diffusivity $K_Z$ (using the Thorpe-scale derived version of the last parameter). The relationship between D and N (Figure 10) is not well-fitted by any standard form of regression line but does have very clear limits. There are no cases where D < 1.15 and N > 0.02 s$^{-1}$, and no cases where N > 0.01 s$^{-1}$ and D < 1. For D > 1.15, there appears to be no relationship between D and N. The maximum D is approximately 1.55 for all N > 0.01 s$^{-1}$, whereas for N < 0.01 s$^{-1}$ it declines sharply with N. Similarly, the minimum D is approximately 1.17 for all N > 0.02 s$^{-1}$, whereas it declines sharply with N for N < 0.02 s$^{-1}$. Data from both lakes follow all these patterns in essentially the same way. Amongst the scale-specific versions of D, the relationship with N is strongest for $D_F$, where a very clear relationship is observed, which is best fitted by an exponential regression line (r = 0.902; *n* = 290; *p* << 0.001 for Esthwaite Water; r = 0.964; *n* = 104; *p* << 0.001 for

Blelham Tarn). This association is weaker, but still evident for $D_I$, although the shape of this plot has more in common with that of D than that of $D_F$ (i.e., no relationship between $D_I$ and N above small values of each parameter, rather than an exponential relationship). The relationship between $D_C$ and N appears essentially non-existent. This implies that the layering structure at fine scales (<20 mm) is strongly associated with the buoyancy frequency (which is calculated as an average value for each two-meter depth bin, so can be thought of as a background or ambient value), but that this association weakens as the scale of the layering increases, so that layering structure at scales of >200 mm has no significant association with the background buoyancy frequency.

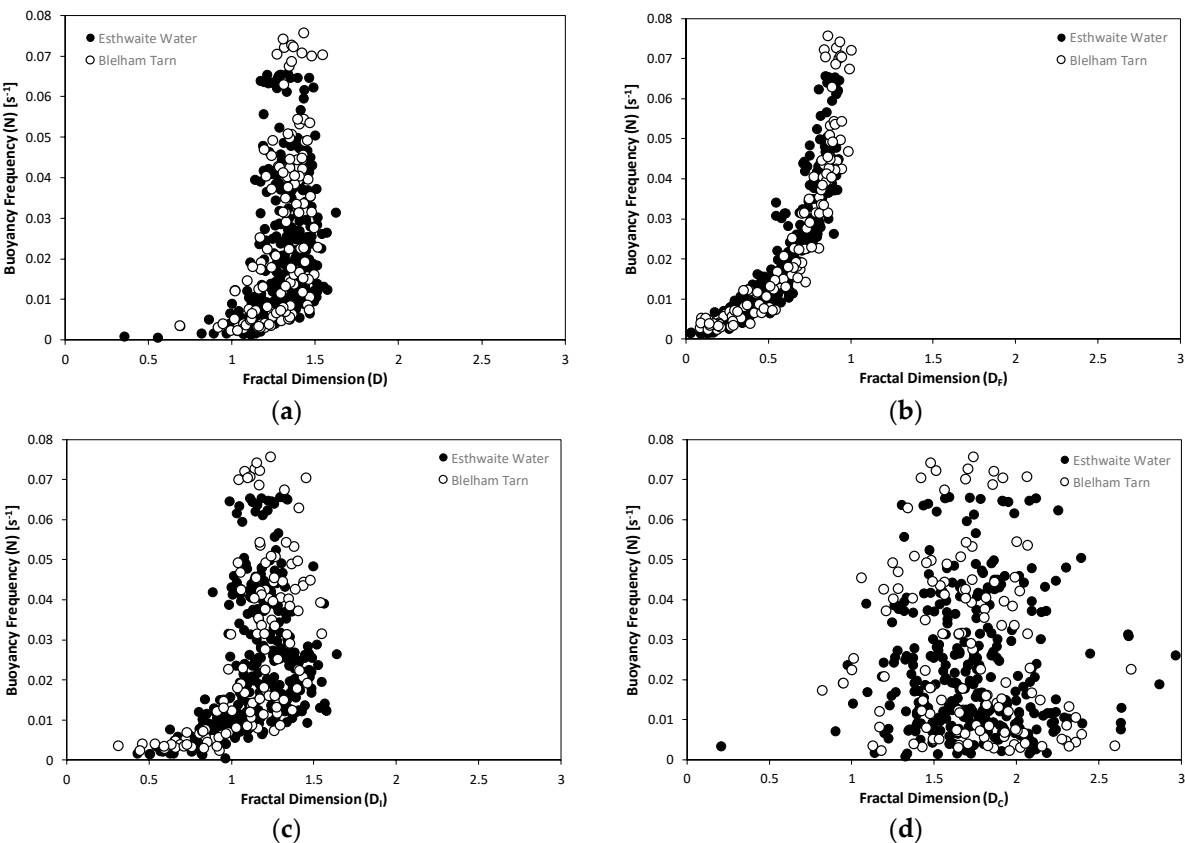

**Figure 10.** Plots of buoyancy frequency against (**a**) D; (**b**) $D_F$; (**c**) $D_I$; and (**d**) $D_C$. Data from all depths bins, all profiles and all dates combined.

The relationships between D (and its scale-specific variants) and the parameters which quantify turbulent stirring and mixing, $L_T$ and (Thorpe scale-derived) $K_Z$ show a consistent pattern (Figures 11 and 12). For both lakes, there is a strongly significant negative linear correlation between D and $\log(L_T)$ (r = −0.533; n = 290; p << 0.001 for Esthwaite Water; r = −0.442; n = 104; p << 0.001 for Blelham Tarn), and D and $\log(K_Z)$ (r = −0.304; n = 290; p << 0.001 for Esthwaite Water; r = −0.298; n = 104; p = 0.002 for Blelham Tarn), with larger D values (more layering structure) corresponding to smaller $L_T$ values and vice versa, but there is also a great deal of scatter around this general trend. Similarly, there is a strongly significant negative linear correlation between D and $\log(K_Z)$ (r = −0.304; n = 290; p << 0.001 for Esthwaite Water; r = −0.298; n = 104; p = 0.002 for Blelham Tarn), implying that larger values of D are found where there is less turbulent diffusion. For both $L_T$ and $K_Z$, the correlation is stronger with $D_F$ (particularly for $K_Z$) and $D_I$ (particularly for $L_T$) (i.e., at scales < 200 mm). As is the case with N, there is no significant association of $L_T$ or $K_Z$ with $D_C$. This implies that the active stirring and mixing processes represented in the turbulent overturns in the temperature profiles are most strongly associated with layering structure in the profile at fine and intermediate scales.

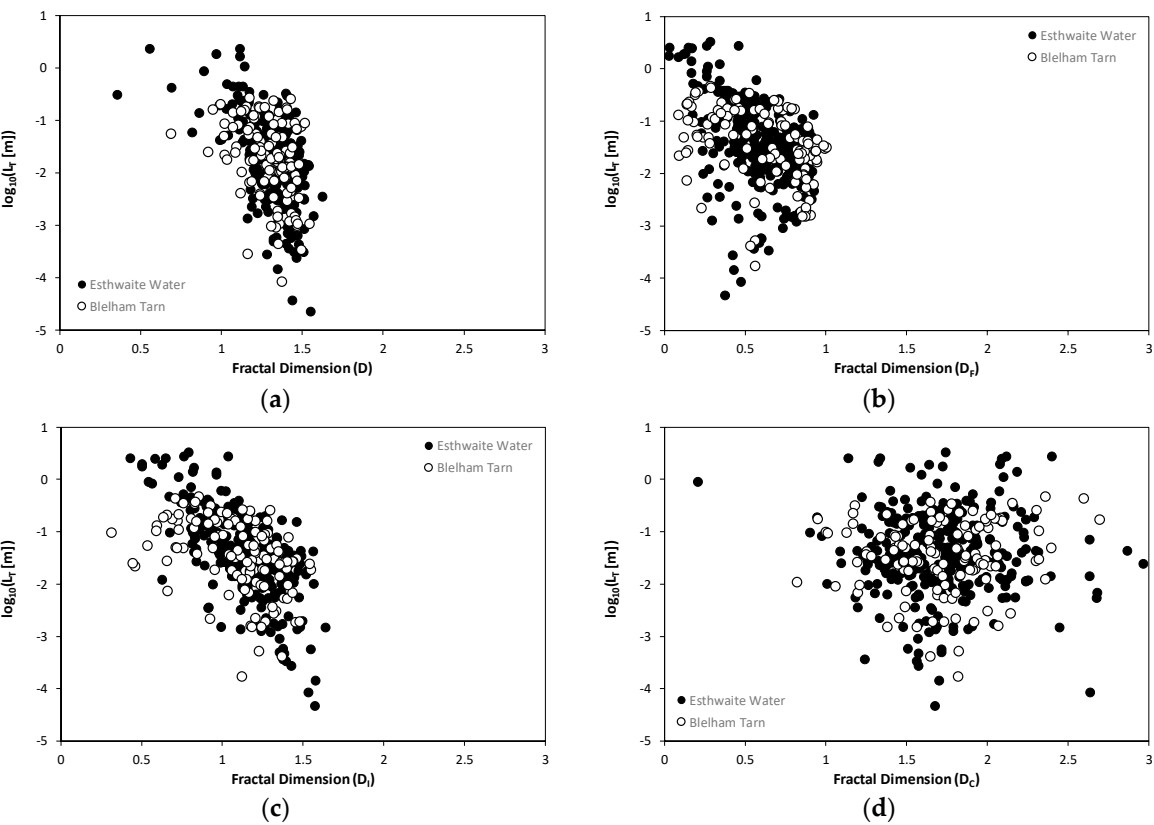

**Figure 11.** Plots of $\log_{10}(L_T)$ against (**a**) D; (**b**) $D_F$; (**c**) $D_I$; and (**d**) $D_C$. Data from all depths bins, all profiles and all dates combined.

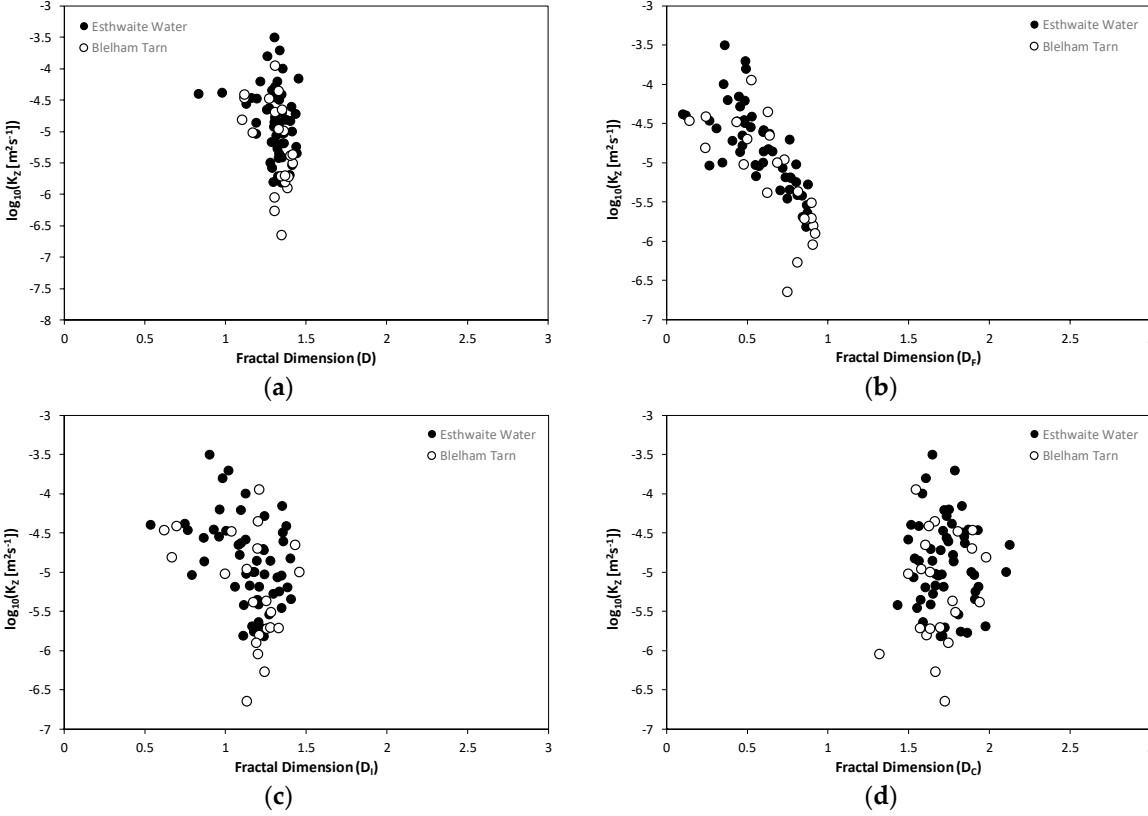

**Figure 12.** Plots of $\log_{10}(K_Z)$, calculated using the Batchelor method, against (**a**) D; (**b**) $D_F$; (**c**) $D_I$; and (**d**) $D_C$. Data from all depths bins, all profiles and all dates combined.

## 4. Discussion

The profiles of temperature and centered Thorpe displacement, together with the mixing layers indicated by the segmentation lines (i.e., the left-hand panels in Figures 5 and 6) show clearly the limited extent of active mixing during the stratified season, its tendency to be most common near the water surface and lake bed, and its increasing prevalence as the stratification breaks down as autumn sets in. This is all very much in agreement with normal expectations for turbulent mixing in lakes [6]. The mixing layer segmentation lines provide a clear visual idea of the proportion of the water column that is actively mixing and the extent to which the mixing layers are separated by non-mixing regions. The changes in the patterns of these aspects of the profiles are conflations of variations on timescales varying from sub-hourly to seasonal, so no attempts have been made to find any consistent trends in them. This is a good example of the way in which collection of microstructure profiles strongly under-samples the temporal variations in turbulence mixing activity.

With regard to the $K_Z$ estimates, the Batchelor method is long-established, and its veracity has been demonstrated in a very wide range of field, laboratory and numerical studies. The Thorpe-scale method, on the other hand, has only been quite recently proposed, and deployed in the analysis of oceanographic, rather than limnological, data [19]. It seems appropriate, therefore, to assume that the Batchelor method values of $K_Z$ are the "correct" ones, and to ask why the Thorpe-scale method values do not equate to them. From the plots in Figures 5 and 6, the two sets of values appear to agree better with each other in the top half of the water column (down to 6 m depth), i.e., above and within the upper part of the metalimnion, than they do below this. Regression of the two sets of values bears this out: for data from <6 m depth (from both lakes, all dates combined), r = 0.662 ($n$ = 144; $p = 1.55 \times 10^{-19}$), whereas for data from below that depth, r = 0.250 ($n$ = 177; $p = 8 \times 10^{-4}$). Moreover, the Thorpe scale method appears incapable of producing values at the larger end of the of values provided by the Batchelor method: for the Thorpe scale method, the range of $\log_{10}(K_Z)$ values across both lakes, and all depths and dates is −6.6 to −4.3, whereas for the Batchelor method, it is −6.6 to −1.5. In the upper 6 m of the water column, the values in the range −4.3 to −1.5 provided by the Batchelor method all occur in the two earliest profiles in the year (22 May and 16 June). These are unusually high values of $K_Z$ for lakes (c.f. typical values quoted by, for example, [6]), but they are not outliers in the profiles from those dates, so there appears no reason to treat them as any less reliable than the other Batchelor method values. Moreover, they occur in regions where stratification is very weak, and are consistent with the intuitive concept that mixing will be rapid in these regions, because it is not very constrained by buoyancy forces.

The Thorpe scale method parameterization of $K_Z$ assumes that conditions are in the energetic regime ($\varepsilon / \nu N^2 > 100$) defined by [20]. This may explain why the values it provides in the deeper part of the water column, below the thermocline, match the Batchelor method values less well than in the upper part of the water column. There is relatively little turbulent mixing in these deeper waters, and the buoyancy frequency is also generally higher than in the upper waters. It is concluded that, in the circumstances studied here, the Thorpe scale method provides a reasonably accurate and relatively straightforward method of estimating $K_Z$, which provides values that compare closely to those provided by the Batchelor method above the thermocline, except at times when the Batchelor method indicates high values of $K_Z$. From the data in this study, "high" in this context means $\log_{10}(K_Z) > -4.3$.

With regard to the pseudo-spectra of layering structure, and specifically the values of the parameter D presented as a convenient quantification of them, the data in Figures 8–10 show that there is a strong relationship between D and the buoyancy frequency N, which persists when the data are averaged across dates and lakes (Figure 8), and when seasonal and daily timescale variations are considered (Figure 9). The plot of $D_F$ vs. N in Figure 10 shows that this relationship is particularly strong when considering the fine-scale layering structure. Conversely, the plot of $D_C$ vs. N shows that buoyancy frequency has no consistent correlation with D at coarse scales. As noted above, the parameters that quantify

turbulent stirring and mixing, $L_T$ and $K_Z$, correlate best with D at intermediate scales, and indicate that there is more layering structure when there is less stirring or mixing.

The fine scale layering structure identified by the analysis presented here and quantified by $D_F$ can be taken as equivalent to the fine-scale structure identified by [28] and to be associated with the dissipation of turbulent kinetic energy ($\varepsilon$), while the coarse scale structure quantified by $D_C$ can be associated with the irreversible changes to the potential energy caused by turbulent mixing (b). The intermediate scale, quantified by $D_I$, is representative of variations that are actively stirring and mixing, and which could go either way—they could break down into finer scale structure and dissipate away, or merge and smooth out into coarser-scale structure and become irreversible changes in potential energy. If $D_F$, $D_I$ and $D_C$ are thought of in this way, Figures 10 and 12 can be interpreted to make a number of suggestions. Firstly, the coarse-scale layering structure (i.e., the vertical distribution of potential energy) recorded in a microstructure profile is essentially unrelated to the turbulent stirring ($L_T$) and mixing ($K_Z$) occurring at the time that the profile was taken (which implies that it is due to previous turbulent activity) and is independent of the average buoyancy frequency (i.e., the amount of coarse-scale layering structure in any given layer is independent of whether that layer as a whole is strongly or weakly stratified). Secondly, the fine-scale layering structure is very strongly correlated with buoyancy frequency (there is more fine-scale structure in more stratified layers) such that there is more fine-scale structure in more stratified layers. It is also correlated with the Thorpe scale and diffusivity coefficient, but less strongly and in a negative sense. This seems counter-intuitive initially—one would expect a fine-scale structure indicative of more turbulent dissipation to be less prevalent in regions of stronger stratification and less turbulent stirring and mixing. Our interpretation of this finding is that it is pre-dominantly a consequence of the fine-scale layering structure being smoothed out quickly in regions of lower stratification, because of the smaller density differences between layers involved, but to be more persistent in more strongly-stratified regions because of the greater density differences between layers. The relationships between the fractal dimension at intermediate scales $D_I$ and N, $L_T$ and $K_Z$ suggest a situation intermediate to those of $D_F$ and $D_C$, indicating that a mix of the drivers determining these relationships at coarse and fine scales are operating at this scale.

The layering structure analysis has the potential to be useful because it analyses an aspect of the data that is indicative of the history of turbulent mixing, not just the mixing that is occurring at the time of the profiling. Ways in which it might be useful are (1) to provide measures of longer term averages of $K_Z$ than current methods of analysing microstructure profilers provide; or (2) to provide alternative estimates of the efficiency of turbulent mixing (i.e., the value of $R_f$, and thus of $\Gamma$) that can be used to triangulate values provided by other methods. To test the first of these, an additional method of measuring the long-term average $K_Z$ is required—for example the temperature diffusion method of [21] or the very similar dye diffusion method used by, for example [38], against which values of D and its scale-specific versions can be assessed. Neither of these were available in this study, so this suggestion remains as a proposal for further study. However, for the sake of providing an indication of what $K_Z$ values this method might provide, the relationships between D and $K_Z$ in Figure 12 are noted. To test (2), and further investigate the extent to which the fractal dimension parameters introduced here can be used to indicate mixing efficiency, requires comparison with results from numerical modelling [13,23].

## 5. Conclusions

The thermal microstructure profiles analysed in this study show the behaviour expected in terms of stratification structure, turbulent mixing activity and vertical variation in the thermal diffusivity coefficient, $K_Z$. While the values of $K_Z$ calculated using two different methods—Batchelor curve fitting to the temperature gradient spectrum; and calculation directly from measurements of N and $L_T$ using the equation of [19] based on the parameterization of $K_Z$ of [20]—show good agreement in many cases, they also

differ strongly in other cases. Given that the Batchelor curve fitting method is very well established and its accuracy has been demonstrated in many previous studies of small lakes, it is recommended that the Thorpe-scale method, whilst attractive for its simplicity and directness, is only used to calculate $K_Z$ values above the thermocline and during the strongly stratified period in mid to late summer in lakes such as those studied here.

The novel analysis of the layering structure and its pseudo-spectra presented here shows that they have some properties that are consistent across the datasets used here, and other properties that vary consistently with other parameters. Values of the parameter D— the slope of the pseudo-spectrum—vary most consistently with the buoyancy frequency, especially $D_F$, the fine-scale specific version of D.

The main limitation of the findings presented here is that, at present, the novel parameter derived, D, has no clear practical use. To address this, it is suggested that D, and its scale-specific variants $D_F$, $D_I$ and $D_C$, as defined here, may be useful in two ways: firstly, to provide measures of longer term averages of $K_Z$ than current methods of analysing microstructure profilers provide; and secondly, to provide alternative estimates of the efficiency of turbulent mixing that can be used to triangulate values provided by other methods. Testing of the ability of these parameters to be of use in these ways requires further work involving field measurements of time-integrated values of $K_Z$ (using, for example dye or thermal diffusion methods) and numerical modelling of turbulent mixing using, for example, DNS methods [13,23].

**Funding:** This research was funded by the UK Natural Environment Research Council, grant numbers NE/F00995X/1 and NE/G010498/1.

**Institutional Review Board Statement:** Not applicable.

**Informed Consent Statement:** Not applicable.

**Data Availability Statement:** The data presented in this study are openly available in Pure at https://www.research.lancs.ac.uk/portal/en/datasets/search.html. (accessed on 26 September 2021).

**Acknowledgments:** I am grateful to Fanghua Li, Rebecca Messham and Eleanor Mackay for their roles in the data collection fieldwork, and to Ian Jones and Joshua Arnott for useful discussions about the data and its analysis.

**Conflicts of Interest:** The author declares no conflict of interest.

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
