# Peer review of "The Multi-Scale Layering-Structure of Thermal Microscale Profiles"

_water, doi:10.3390/w13213042_

Round 1

Reviewer 1 Report

Please see the section-wise comments below on your research article with the following details.

Manuscript title: The multi-scale layering-structure of thermal microscale profiles

Manuscript Number: water-1393587

Journal Submitted: Water

Specific Comments:

Title:

The title is well placed and represents the study.

Abstract:

First of all, it is not suitable to use “I” repeatedly. You may use alternatives.

Kz is not defined.

The abstract is written excellently.

However, the keywords must be written alphabetically.

Introduction:

Figure 1. Please designate each part of the figure as a,b, and c and mention their details in the figure caption.

L 74: What is the reason for this name?

Please mention and define your aims clearly and not as a general sentence. You must specify.

L 134-139: This must be either deleted or moved here to a more suitable place.

Materials and Methods:

L 155-162: I could not see any background references.

Figure 2. Increase the font size of figure axis labels, move the figure numbers (a, b, c, and d) inside the figures, and increase their sizes by moving them closer to each other to enhance the visibility.

L 270-271: This must be deleted.

Figure 3. Same as for figure 2.

Figure 4. Excellent piece.

Where are the statistical descriptions? Please provide all the details.

Results:

Figures 5 and 6. The distances between each figure (sub-figures) are too much. Please try to rearrange and decrease the length. Instead, you may enhance the figure size to increase visibility.

Figure 7. Please avoid using # sign in the axis label.

Figure 8. You have used two different brackets in two different axis labels.

Figure 9. The small figures on the right side are too small and invisible. Please work on it and increase the clarity.

The description of the results is excellent and provided with all the details. Excellent job.

Discussion:

I have great applaud for the author for conducting this section so nicely and excellently.

Conclusions

Could you please provide the main limitations of this method?

L 616-623: I guess this can be moved to the discussions section.

References:

There are several studies on this subject; however, you have not cited many of them. Please substantiate your work with more of the citations from previous literature.

Concluding remarks:

I must say that this is an excellent article, and it shows the skill of the author. Great job.

Author Response

(Reviewer's comment in bold, my responses in normal font)

Title: The title is well placed and represents the study.

Thankyou, no response required.

Abstract: First of all, it is not suitable to use “I” repeatedly. You may use alternatives.

I would argue that these days, this is a stylistic choice - I have seen many published papers that use "I" repeatedly. Nevertheless, I respect the point being made and have altered the abstract to remove all uses of "I".

Kz is not defined.

Kz is named (defined) as the coefficient of vertical turbulent diffusivity at lines 14-15. I believe that any more detailed definition than this would not be appropriate in an abstract. It is more fully defined, and methods for calculating it are detailed at lines 70-99 in the main text.

The keywords must be written alphabetically.

Corrected

Introduction:

Figure 1. Please designate each part of the figure as a,b, and c and mention their details in the figure caption.

Done.

L 74: What is the reason for this name?

I assume the name being asked about is the "Batchelor method"? This is so-called because it was first formulated by G.K. Batchelor, as detailed in reference [17]. I have changed the wording here slightly to make this link explicitly. 

Please mention and define your aims clearly and not as a general sentence. You must specify.

Text adapted at lines 134-145 to address this.

L 134-139: This must be either deleted or moved here to a more suitable place.

This has been achieved by the changes made at lines 134-145 in response to the previous point.

Materials and Methods:

L 155-162: I could not see any background references.

I'm not sure any are necessary here - this paragraph just describes the operation of the instrument used. However, for completeness, I have added a reference [33], and updated the reference numbering accordingly.

Figure 2. Increase the font size of figure axis labels, move the figure numbers (a, b, c, and d) inside the figures, and increase their sizes by moving them closer to each other to enhance the visibility.

I followed the formatting shown in the Frontiers template document for all of the figures, which has the letters outside of the figures, so I have not changed those. My view is that increasing the size of the figure axis labels would be disproportionate with the size of the plots themselves, so have left these as they were.

L 270-271: This must be deleted.

Removed

Figure 3. Same as for figure 2.

Please see my response to the comment about Figure 2 above.

Figure 4. Excellent piece.

Thank you!

Where are the statistical descriptions? Please provide all the details.

Text added at lines 270-275 to address this.

Results:

Figures 5 and 6. The distances between each figure (sub-figures) are too much. Please try to rearrange and decrease the length. Instead, you may enhance the figure size to increase visibility.

As above, I've followed the Word template and instructions on figure formatting provided to arrive at this arrangement, so I have not changed these. If the editors wish, I can adjust them to address the reviewer's comments, but would seek their guidance about how far I can stray from the template instructions. 

Figure 7. Please avoid using # sign in the axis label.

Changed to "Number"

Figure 8. You have used two different brackets in two different axis labels.

Yes, the (D) is for an abbreviation, whereas the [m] is for units (metres). 

Figure 9. The small figures on the right side are too small and invisible. Please work on it and increase the clarity.

I don't feel that these are worth increasing in size, as they provide only supplementary information. Given that this is primarily an online journal, so readers can zoom in to see these images, I thought that their size was acceptable (the quality is retained on zooming in so that they are easily readable). If the editors' view is that they are too small, I would happily remove them, but I have kept them in at this point in case they are acceptable.

The description of the results is excellent and provided with all the details. Excellent job.

Thank you!

Discussion:

I have great applaud for the author for conducting this section so nicely and excellently.

Thank you!

Conclusions

Could you please provide the main limitations of this method?

A sentence added at 627 to address this point.

L 616-623: I guess this can be moved to the discussions section.

It could be, but because it links with the limitation comment added at 627, and points the way forward from this study, I feel it goes best in the Conclusions, and as an appropriate way of finishing the text.

References:

There are several studies on this subject; however, you have not cited many of them. Please substantiate your work with more of the citations from previous literature.

There are now 38 sources cited, most of which are directly on this topic. My view is that this is a more than appropriate number of citations for a paper of this size. I could add more references if the editors agree with the reviewer on this point, but I feel that this would mainly just be for the sake of having more references.

Concluding remarks:

I must say that this is an excellent article, and it shows the skill of the author. Great job.

Thank you!

Reviewer 2 Report

It is a classical subject but still carries its weight. So, it is good to see some new development. However, more explanations are required as below.

  1. The de-noising method employed in this study was originally used in the ocean. Could it be applied to freshwater without any modification?
  2. How 1 mm resolution can be justified?
  3. Under what meteorological conditions, the analyzed profiles were recorded? Please explain why it is not necessary to bring wind-induced shear stress into the analysis of turbulent mixing?
  4. How do the conclusions such as the two methods show good agreement in many cases, they also differ strongly in other cases, are related to multi-scale layering structure? The value of introducing the multi-scale layering structure should be better concluded.

Author Response

It is a classical subject but still carries its weight. So, it is good to see some new development. However, more explanations are required as below.

Thank you for your positive review and helpful and insightful comments. 

The de-noising method employed in this study was originally used in the ocean. Could it be applied to freshwater without any modification?

This is an interesting point, but on reflection, I don't see why it can't be applied to freshwater without modification. Turbulence behaves very much the same in fresh and salt water, and responds to density stratification in a similar way in both cases. The theory of turbulence in natural waters has been developed historically with a mix of results from fresh and salt water contexts, so it seems to me to be appropriate to take a method derived in salt water and apply it to freshwater (as has been done by many others in the past) as I have done here.

No changes made to manuscript in response to this comment.

How 1 mm resolution can be justified?

The SCAMP instrument records data at 100 Hz and moves vertically through the water column at approximately 10 cm per second, which implies a spatial resolution of its recorded data points of 1 mm. This is explained in the text at lines 166-167 in the revised manuscript. Thus, it seems appropriate to interpolate to this spatial scale - neither creating a pseudo-higher resolution nor losing data by going for a lower resolution. I've added a few words at lines 166-167 to emphasize the explanation made there.

Under what meteorological conditions, the analyzed profiles were recorded? Please explain why it is not necessary to bring wind-induced shear stress into the analysis of turbulent mixing?

The meteorological conditions were varied - some days were sunny and calm, other cloudy or with some wind. Both sites (Esthwaite and Blelham) are reasonably open to the wind (Esthwaite more than Blelham). However, I was not intending to analyze the causes of the turbulent mixing (which would no doubt have included wind-induced shear stress) in this paper. The paper is focussed on the details of the structure of the stratification of the water column and its relationship to N, Kz and LT (all of which will have been affected by wind stress, but going into the causes of the turbulence and stratification was not part of the aims of the paper). 

No changes made to the text in response to this comment. 

How do the conclusions such as the two methods show good agreement in many cases, they also differ strongly in other cases, are related to multi-scale layering structure? The value of introducing the multi-scale layering structure should be better concluded.

Yes, I would have liked to have gone further with concluding about the value of introducing the multi scale layering structure, and the D parameter I introduce. However, the data (or, at least, my analyses of it) did not allow this and I was unable to identify the sorts of relationships between the conclusions that the reviewer (understandably) suggests. I have therefore tried to point the way forward so that other researchers (and/or myself in future work) might be able to address this sort of point. 

No changes made to text in response to this comment.